# Approach to Small Animal Neurorehabilitation by Locomotor Training: An Update

**DOI:** 10.3390/ani12243582

**Published:** 2022-12-18

**Authors:** Débora Gouveia, Ana Cardoso, Carla Carvalho, António Almeida, Óscar Gamboa, António Ferreira, Ângela Martins

**Affiliations:** 1Arrábida Veterinary Hospital—Arrábida Animal Rehabilitation Center, 2925-538 Setubal, Portugal; 2Superior School of Health, Protection and Animal Welfare, Polytechnic Institute of Lusophony, Campo Grande, 1950-396 Lisboa, Portugal; 3Faculty of Veterinary Medicine, Lusófona University, Campo Grande, 1749-024 Lisboa, Portugal; 4Faculty of Veterinary Medicine, University of Lisbon, 1300-477 Lisboa, Portugal; 5CIISA—Centro Interdisciplinar-Investigaçāo em Saúde Animal, Faculdade de Medicina Veterinária, Av. Universi dade Técnica de Lisboa, 1300-477 Lisboa, Portugal

**Keywords:** locomotor training, treadmill, spinal cord injury, neurorehabilitation, dogs, cats

## Abstract

**Simple Summary:**

Locomotor training (LT) is a main strategy for functional neurorehabilitation that can promote stepping relearning, leading to a coordinated and modulated ambulation pattern, and can be applied to a wide range of neurological diseases, namely after spinal cord injury. This review article offers a neurophysiological explanation and an overview of current evidence on LT. Additionally, it provides the knowledge on how to perform this exercise through land and underwater treadmill training with primary guidelines and feasible protocols for small animals that may be meaningful on a day-to-day basis for implementation in the veterinary setting.

**Abstract:**

Neurorehabilitation has a wide range of therapies to achieve neural regeneration, reorganization, and repair (e.g., axon regeneration, remyelination, and restoration of spinal circuits and networks) to achieve ambulation for dogs and cats, especially for grade 1 (modified Frankel scale) with signs of spinal shock or grade 0 (deep pain negative), similar to humans classified with ASIA A lesions. This review aims to explain what locomotor training is, its importance, its feasibility within a clinical setting, and some possible protocols for motor recovery, achieving ambulation with coordinated and modulated movements. In addition, it cites some of the primary key points that must be present in the daily lives of veterinarians or rehabilitation nurses. These can be the guidelines to improve this exciting exercise necessary to achieve ambulation with quality of life. However, more research is essential in the future years.

## 1. Introduction

Functional neurorehabilitation (FNR) is based on neuroanatomy and neurophysiology principles, allowing the development of intensive protocols that are implemented in human medicine [1]. It is mandatory to understand the function of spinal cord interneural circuits and the neural mechanisms necessary for locomotion control, which is the only way to support the progress of new targeted locomotor rehabilitation strategies. Thus, the physiological knowledge in animals and humans allows the optimization and evolution of neurorehabilitation protocols [1,2].

The evidence of signal transmission through a spinal cord injury (SCI), both caudally and rostrally to it, was the starting point for continuous stimulation of the remained and residual central axons pathways [3,4,5]. Experiments in animal models, such as cats with transected spinal cord, showed that treadmill training was effective to promote gate restoration [6]. Thus, functional training may promote the reorganization of neural locomotor networks after the loss of motor ability and following neuronal injury [2,7]. 

One of the main strategies for this is locomotor training (LT) because it could promote multi-stimulation from different manners, remaining the question of how to perform this training in a beneficial and safe way. An accurate LT may promote postural control and postural standing ability (assisted or not) by a neuromuscular organization that depends on spinal cord changes, usually biochemical alterations [8,9]. 

These exercises allow the dynamic interaction between afferent inputs from all functioning receptors, such as proprioceptive and biomechanical, located on: the intrafusal fibers of the muscle and the Golgi tendinous organ (muscle fibers group Ia and Ib) [10]; hip joint (mechanoreceptors); other joints and skin (stretch and sensitive mechanoreceptors) [11].

The LT recruits and adjusts the number of motoneurons that generate rhythmic movements and, at the same time, stimulates the peripheral receptors that provide nearly 30% of motor output [1].

The central pattern generators (CPG), located in the lumbar segments of the spinal cord [12,13,14,15], may be triggered by external stimuli provided by the treadmill belt movement, which is a common exercise performed in complete SCI dogs, cats, and rats to achieve normal stepping [6,16,17,18,19,20].

Therefore, one tool that provides this type of exercise is treadmill training, which allows stepping relearning, depending on the exercise’s practice. The performance success is highly variable, a fundamental feature of the neural movement control [21] based on plasticity improvement on spinal neuronal circuits, already established in different studies with “complete” SCI cats [6,21,22,23,24,25].

Those quadrupedal animals exhibited clear evidence that the spinal network, grounded on the CPG and motor interneurons, may be the start of generating locomotor rhythm [26]. Moreover, in human medicine, active assisted movements are studied concerning their potential benefit on motor control and spasticity [27]. 

Thus, treadmill LT allows the neuronal mechanism stimulation of the spinal segments caudally to the injury by stimulus of serotonin and/or noradrenergic pathways, promoting neural plasticity. This phenomenon potentiates spinal reflex excitability, which in part relies on glutamate action at N-methyl-d-aspartate (NDMA) receptors and brain-derived neurotrophic factor acting at TrKB receptors, possibly modulated by the treadmill training [28]. 

The main objective of this review was to obtain a neurophysiological explanation for locomotor training. In addition, address the applicability and primary guidelines for the clinical setting. Finally, accomplish protocols for possible motor recovery with coordinated and modulated movements necessary for ambulation. 

### 1.1. Body-Weight-Supported Treadmill Training

Body-weight supported treadmill training (BWSTT), which usually refers to 60% to 80% of supported weight, is a method that allows activity-dependent plasticity enhancement and the residual pathways and networks [29,30,31,32]. In particular, the BWSTT aims to improve coordination and ambulation, focusing on the lumbar locomotor CPG circuits [8,33,34,35,36,37,38].

In human patients, these BWSTT exercises have been shown to reduce spasticity by replacing abnormal hyper-excitable sensory firing with functional afferent signaling, which decreases muscle spasm and co-contraction [39,40]. Moreover, the LT repetitive movements potentially improve glutamatergic dorsal root ganglia (DRG) sensory feedback, contributing to the primary extrinsic source stimulation and entering the spinal cord below the injury to engage the local circuits, mechanisms already evidenced in animal models [41].

### 1.2. Proprioceptive Sensory System

It is essential to consider the muscle spindle work, which is innervated by proprioceptive sensory neurons and conducts the muscle contraction information to the spinal cord. They are the proprioceptors that exhibit the most widespread central projection of all DRG sensory neurons and establish synaptic connections with motor neurons and interneurons of the intrinsic circuits [42]. Therefore, the muscle spindle afferent is mainly the key to the direct excitation of spinal circuits needed for motor control upregulation, mostly under disconnected descending input situations [43].

The cholinergic propriospinal cells significant for the CPG control and the coordinated locomotor output may be stimulated by the LT, contributing to the recovery of locomotion in the absence of descending locomotor control [44]. Thus, descending propriospinal neurons may provide neural networks excitation, regardless of supraspinal control. This activation of interneural networks could potentially promote locomotor coordination and is an effective strategy for stepping recovery [45,46] (Table 1).

## 2. When to Perform the Locomotor Training?

The time to initiate LT is also of primary concern. Early treadmill training after severe SCI has been shown to promote the possibility of an increase in muscle electromyography signal and modulate activity over the step cycle, similar to healthy human patients. Furthermore, surface electromyography findings are shown to correlate with motor recovery [47,48]. 

There is a diminished noradrenergic input to neural networks in SCI, decreasing signal amplitude [49]. Moreover, dendritic persistent inward currents (PICs) and the motoneuron dendrites activation [50,51,52,53] may be restored with monoamines, which are released during exercise and have several effects on motoneurons and the PICs [54,55]. 

Post-operative rehabilitation may be safely used in the first 24 h after surgery, provided it does not worsen spinal hyperesthesia or decrease the neurological status [19,56,57]. Also, moderate-level evidence research has supported the implementation of LT within the first three days after surgery, in contrast with activity restriction (e.g., cage rest) that was supported by low-level evidence, demonstrating that this restriction does not mean annulment of LT and rehabilitation exercises [56] (Table 2). 

Thus, automated locomotor training is an essential tool already implemented in human patients with severe motor impairment. The choice of the device to assist training depends on the sensorimotor deficits and the cardiopulmonary restrictions [58], leading to a bodyweight-bearing training and a high repetitive rhythmic pattern with the help of a body harness [59]. In SCI human patients, this exercise can be performed through manual assistance by therapists or, mostly in cases of severe deficits and cardiorespiratory restrains, continued by robotic exoskeleton assistance or robotic assisted treadmill training (e.g., Lokomat^®^) [60,61]. On the other hand, in the veterinary field, spinalized animals presented good results when performing treadmill training [18,19,20,62,63], also in comparison with walking on different floors [64]. The alternative to robotic assistance in small animals that require body weight support has been the use of continuous wheelchair training, or in heavier dogs through a passive standing device above the treadmill [62]. 

Long-term locomotor training has been correlated to promote bone and muscle mass increase along with positive cardiovascular effects [45,65,66,67,68], intralimb hip-knee coordination [69], activation of propriospinal pathways that mediate interlimb coordination [70], improve muscle force output and endurance [71].

### Repetitions Number

The number of repetitions and the functional consequences of locomotor training on neuronal plasticity is motor learning principles that must be implemented to increase performance [8] (Table 3). 

Research studies have shown that most known protocols, particularly 82%, are based on one daily training session [72,73,74,75,76,77], and only 18% were based on two to three daily sessions [78,79,80]. In the same studies, the duration of each session may vary, from 5–15 min (53%) to 20–30 min (47%), and the most frequent exercise implemented was treadmill training (33%), followed by the BWSTT (17%). The total rehabilitation time varies, from 1–36 weeks, with 15 studies reporting training five days/week [73,77,81,82,83,84]. The specific amount of time needed for standing or stepping also depends on each subject’s level of exercise-induced fatigue [67]. 

In translation for small animals, there are few reports of rehabilitation sessions manly with underwater treadmill locomotor training [85,86]. Recently some studies have been published describing locomotor exercises on land and underwater treadmill, in acute and chronic intervertebral disc disease (IVDD) dogs [19,20], SCI contusion cats, cervical IVDD dogs and in acute non-compressive nucleus pulposus extrusion (ANNPE) dogs [62,63]. 

In human patients, intensive rehabilitation therapy was performed five times/week for 90 min sessions, with step training for a minimum of 20 min and long-term follow-up evaluations for 6–12 months [9]. After intensive training, motor recovery in incomplete motor SCI patients was registered within the first two months of rehabilitation [87]. 

## 3. How to Perform the Locomotor Training?

The locomotor training implementation depends on the cause of the problem and has to consider the strict criteria of each situation. For example, in rehabilitation following SCI, it is mandatory to ensure stabilization of the spinal cord, performing the exercises without oscillations of the vertebral column. 

Thus, the application of treadmill training is based on the possibility of guaranteeing the step cycle repetition in an easier and faster way. If the patient has monoplegia/monoparesis, paraplegia/paraparesis, or tetraplegia/tetraparesis, the adaptation to the treadmill surface is necessary to obtain voluntary or automatic movement. For that, the surrounding environment must be calm and quiet with classical music [19,20,62,63,88], but also with enough energy to promote motivation to start the step cycle, autonomously or with assisted exercises, such as the established bicycle movements, that may require perineal or tail stimulation [18,64,89].

Regarding bicycle movements, and in the case of hypotonic muscles, they should be performed with stretching limbs and vigorous stimulation of cutaneous afferent receptors on the treadmill surface [63]. However, if the dog or cat demonstrates hyperreflexia and increased muscle tonus, bicycle movements have to be smoother, and it is not advised the stretching of intrafusal fibers group Ia/Ib. 

### Locomotor Exercises

For neurologic dogs after SCI, the LT initiation may start with passive kinesiotherapy exercises for adaptation to the treadmill belt, essentially in deep pain negative (DPN) patients, resorting to the BWSTT step training (Figure 1). Next, on the bipedal treadmill training, the forelimbs remain stationary on a fixed platform above the belt [89]. At the same time, the perineal area is stimulated by suspending and crimping the tail or with assisted bicycle hindlimbs movements [5,19] (Figure 1A).

For each training session, variables such as the walking speed and duration may start from 0.8 km/h (0.5 mph) to a maximum of 1.9 km/h (1.2 mph) [71,90,91], over 5 min (4–6 times/day, six days/week), until achieving 20 min (2 times/day, six days/week) [92] (Table 4).

After quadrupedal training starts, patients may receive similar stimulation regarding speed and frequency, aiming to achieve 30–40 min (2 to 3 times/day, six days/week) [93]. Moreover, the treadmill slope should be elevated from 10° [70] to 25° [42] to encourage forelimb–hindlimb coordination [94] (Figure 1B). 

Patients should perform quadrupedal training even in an early stage for complete stimulation, mainly of the residual descending pathways. This training allows a possibility of greater stimulation of the propriospinal system and the brain–brain stem-pelvic limb loop stimulation [18].

The previous guidelines should be implemented in patients with acute post-surgical compressive myelopathy, where patients should be managed with stabilization of the vertebral column and minimum oscillations. The standard protocol is the same for the same type of patient but in a chronic situation. However, the primary kind of exercise should be quadrupedal-step training. 

Furthermore, the underwater treadmill (UWTM) (Figure 2) training could be applied from the second day of admission, with water temperature ~26 °C [95], beginning in 5 min until one hour per day (5 days/week) and speed from 1 km/h (0.28 m/s) to 3.5 km/h (0.97 m/s) [49,96], always looking for signs of overtraining. With regard to the UWTM, patients should have a rest of 48 h, due to the increase of volume of oxygen (VO_2_), heart and respiratory effort, and body energy consumption, which is higher compared to walking on land [97].

In chronic or acute SCI patients, LT performed on the UWTM should be initiated as early as possible. Different protocols are described in Table 4, considering each disease, post-operative or conservative management option and disease progression stage [19,56]. All guidelines are dependent on the cardiovascular and motor ability of each patient, and training should not be performed without monitorization of vital parameters (mucous membranes; capillary refill time, heart rate, respiratory rate; blood pressure, electrocardiogram, muscle fasciculations, spinal hyperesthesia and neurological grade), especially in intensive exercise with increasing time, speed, and slope (Figure 3). The same approach could be applied in patients with degenerative myelopathy and geriatric vestibular syndrome. 

The many variables associated with the training protocol implementing exercise to any patient a challenge, differences in intensity, duration, pattern and frequency of training sessions, and the time post-injury when the exercise begins are of great importance [98].

## 4. Muscle Fatigue

Remple and colleagues (2001) [99] suggested that LT has the potential to induce changes in neural function within the spinal cord, promoting increased motor unit recruitment and resistance training, which increases excitability potential [100,101] and recruitment of spinal motor neurons [101,102,103].

In human patients, motor training was increased over six weeks until stimulation could be maintained for 30 min, similar to what may be achieved in dogs and cats, except adding 1 kg of weight bearing once the 30 min could be performed without fatigue at a given resistance [104]. The effects of leg weights, when adding 1% to 2% of the dog’s total body weight depending on the limb strength and recovery stage, at the level of the carpus/tarsus, are related to the muscle activity increase in the back muscles, promoting stabilization of the vertebral column, particularly when the contralateral limb is in the swing phase [105,106,107]. These types of eccentric exercises may also reduce intracortical inhibition and increase corticospinal excitability by 37–51% [108]. 

Moreover, in humans, fatigue resistance has been shown to improve significantly after only three months of exercising, in agreement with other studies [109,110,111,112], probably due to the LT contribution in improving oxidative and glycolytic enzymes metabolism [113,114] (Figure 4A,B). 

Thus, it is likely that the longer the daily training duration, the more fatigue resistance improves, achieving better results than previously reported, which could be associated with a type II to type I fiber conversion [104]. To Harness and colleagues (2008) [115], this type of approach was an intervention that might be useful to improve impairment and disability after SCI as a primary or adjuvant treatment [116,117]. Furthermore, some studies in dogs have reported the success on using the treadmill running as a modality of choice [118,119,120,121]. 

Dogs and cats that are DPN are the primary targets for this treatment strategy (Table 5), which may increase the level of motor pool activation and promote modulation [122] by performing steps backwards when the treadmill belt direction is reversed [123] (Figure 5A). 

One other training exercise that can be introduced is based on walking stimulation on a land treadmill with step-over obstacles attached to the belt (Figure 5B). This approach was previously presented in a series of experiments in cats with severe motor control deficits, suggesting that both corticospinal and rubrospinal pathways were damaged, and supported by the fact that the properties of neurons in both the motor cortex and red nucleus are compatible with modifications of gait, which are needed to step over obstacles [124].

Furthermore, human medicine research based on constraint-induced or forced-use training and treadmill training, with a particular interest in stroke and SCI patients, has allowed the development of a new field of rehabilitative restoration strategies that goes beyond task-specific training to include optimization of functional recovery through mechanisms of plasticity and regeneration [125].

## 5. Anti-Inflammatory Role

Locomotor training for motor recovery had a growing consensus among neuroscientists that plasticity and regeneration are not limited to the acute phase of injury and may also occur through the chronic phase [126,127]. The inflammatory process increases the release of nitric oxide due to the physiological role of neuronal nitric oxide synthase (nNOS) and endothelial NOS (eNOS), which produces nitric oxide following Ca^2+^ influx [128], causing tissue damage. Thus, a decrease in the inflammatory phase may contribute to more regeneration.

Treadmill exercises exerted their beneficial effects via a significant reduction of C-reactive protein and antioxidant ability [129,130], allowing a decrease in inflammation and restoration of anti-nociceptive inhibitory process, which suggests LT as a powerful exercise for management of neuropathic pain. There are evidence that the secondary anatomical and physical alterations may decrease inflammatory response and increase neutrophin levels, promoting neural regeneration [131]. In rats, it is hypothesized that early moderated exercise may be a therapeutic strategy for non-brain circulation, neuro-inflammation, and astrocytic coverage of brain vessels [132].

Moreover, the inflammatory disorder engages macrophages and/or perivascular cells that seem to play an essential role in producing nitric oxide via the inducible (iNOS) gene, expressed by microglial cells, increasing cytotoxic effect that was previously reported in association with the spinal cord cavitation phenomenon [128]. In the chronic phase, initial tissue necrosis develops into cavity formation, axonotomy, axonal demyelination, glial activation, and scarring [133]. 

Consequently, LT enhances the increase of M2 macrophages that promote angiogenesis [134,135,136] and matrix remodeling while suppressing destructive immunity, promoting functional recovery after SCI [137] depending on the volume of such cavities that can be estimated with 3D imaging [133,138]. In addition, the metabolic ability of muscle changes, measured by oxidative enzyme activity and concentrations of Na+/K+ ATPase, improves with LT, reducing fatigability after SCI in humans and rats. Thus, LT with or without electrical stimulation is likely required to improve muscle endurance via increasing muscle oxidative ability [114]. 

Findings also report the favorable effect of exercise in reducing the risk of neuroinflammatory disorders [139] possibly linked to the decrease of anti-inflammatory cytokines, such as IL1*β*, IL6, and TNF*α* [140], and adipokines via the muscle-adipose crosstalk [130,141,142]. 

LT is an example of intensive training based on PICs and voltage-sensitive modulation. The PICs require concomitant activation of serotonergic (5-HT) and noradrenergic (NA) receptors located on the motoneurons, which are modulated by 5-HT (2B/C) receptors specifically. The role of 5-HT and NA receptors in facilitating motoneurons PICs was first demonstrated in the decerebrate cat [143], but further studies are needed.

## 6. Regenerative Role

There are evidence that anatomical and physical changes may occur after LT, promoting neural regeneration [131]. Several studies in complete/incomplete SCI cats have shown step training on the treadmill [23,144,145] and a consequent increase in neurotrophin delivery [131]. 

Within the neurotrophins, the brain-derived neurotrophic factor (BDNF) plays a vital role in reorganizing the central nervous system as a potent neuromodulator for neural and nociceptive regeneration [59,146]. Moreover, the nerve growth factor (NGF), neurotrophin–3 (NT-3), and neurotrophin-4 (NT-4) are crucial for axonal growth and the remyelination of demyelinated axons, aiming to improve signal conduction across the injury site. Thus, LT may contribute to the disinhibition of the descending motor pathways inhibited with SCI, stimulating the pre-motor neuronal control and promoting intralimb and interlimb coordination [146,147,148]. 

Repeated cyclic exercises over time, every day and for several weeks, have been described to promote spinal reflex locomotion [18,19,20,149]. This approach allows plasticity in the reflex pathways [42,150], and it is known that limited residual descending inputs may be needed for significant functional improvements [9]. Therefore, motor training with a monotonous repetition of the same sensorimotor exercise could result in spinal learning and memorization [30]. 

Locomotor training helps to activate the substantial population of long ascending and descending propriospinal interneurons that connects the cervical and lumbar enlargements via the ventrolateral funiculus, contributing significantly to the neural coupling between cervical and lumbar spinal segments [1,46]; further research is needed. 

For LT monitoring, it should be essential to measure the biomarkers concentrations, such as the glial fibrillary acid protein (GFAP) and phosphorylated neurofilament heavy chain (pNFH) [19,46,151], at the beginning of protocol and throughout the outcomes evaluations, which would allow early identification of progressive neuronal damage [19,152]. 

## 7. Conclusions

For Barrett et al. (2013) [153], neurorehabilitation has a wide range of treatments to achieve neural regeneration, repair, and dynamic reorganization of functional neural systems based on learning experience and neurophysiological stimulation. LT is essential to successful neurorehabilitation for veterinarians, nurses, and technicians, with sharp differences in evolution that are felt or observed every day/week. Thus, the performance of the LT should be different each day, with significant changes in gait patterns that can be observed in just one session. Research must be continued and staff education is necessary, promoting LT as an exciting, innovative, and not boring exercise.

In addition, early intensive LT should be associated with multimodal modalities [18,19,20,63,85,88,123] in all neurological dogs and cats with or without pain perception after surgery or with conservative management, maintaining the stability of the SC. This early and intensive multimodal training can be applied across a diverse range of neural diseases, promoting research for years to come.

## Figures and Tables

**Figure 1 animals-12-03582-f001:**
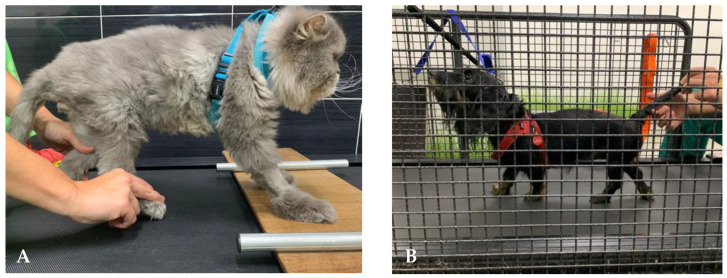
Land treadmill bipedal step training on a cat (**A**) and quadrupedal step training on a dog (**B**).

**Figure 2 animals-12-03582-f002:**
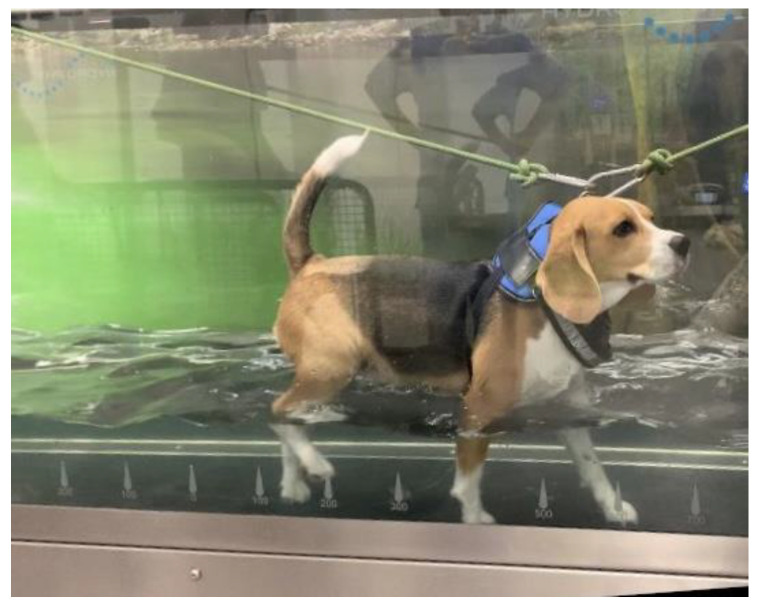
Underwater treadmill quadrupedal training on a dog.

**Figure 3 animals-12-03582-f003:**
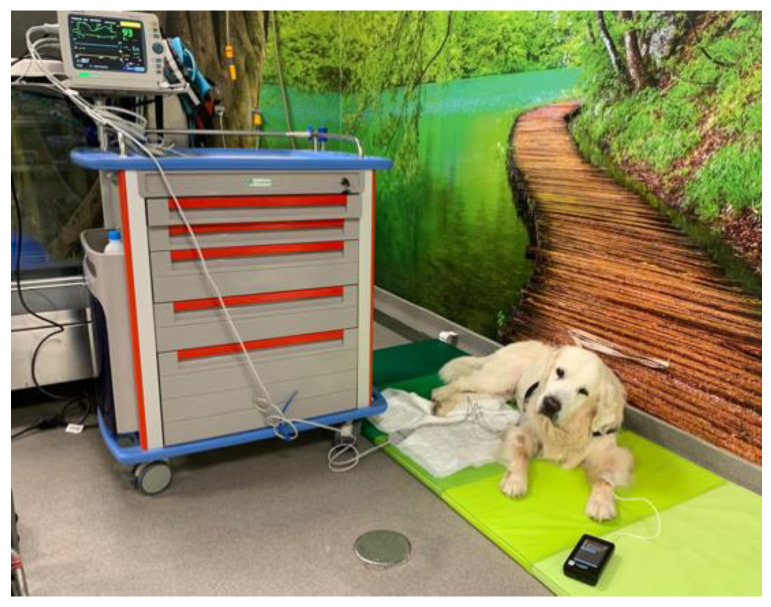
Monitorization of a dog after the performance of locomotor training.

**Figure 4 animals-12-03582-f004:**
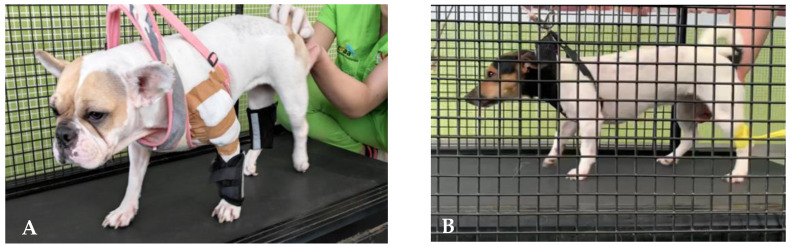
Land treadmill training in an SCI dog with the addition of weight-bearing pads (**A**) and with thera-band resistance (**B**).

**Figure 5 animals-12-03582-f005:**
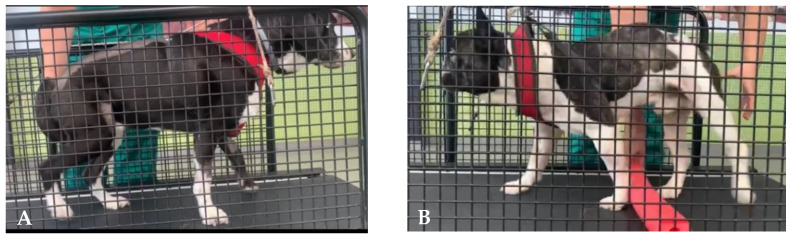
Land treadmill training with a dog performing steps backwards (**A**) and stepping over obstacles (**B**).

**Table 1 animals-12-03582-t001:** Key point n° 1—The contribution of the proprioceptive system for locomotor training.

**Keypoint n° 1:** *“Locomotor training is an extrinsic source of stimulation that may improve glutamatergic input, promoting the activation of intrafusal muscle fibres, proprioceptors, cholinergic propriospinal cells and interneural networks.”*

**Table 2 animals-12-03582-t002:** Key point n° 2—Importance of early implementation of locomotor training.

**Keypoint n° 2:** *“Locomotor training is safe and may be implemented at least until 3 days after surgery or even 24 h post-injury in conservative treatment.”*

**Table 3 animals-12-03582-t003:** Key point n° 3—Exercise repetitions contribution to locomotor training.

**Keypoint n° 3:** *“The outcome of neurorehabilitation depends on the type, number and time duration of repetitions. Also, on the quality of the motor functions.”*

**Table 4 animals-12-03582-t004:** Locomotor training protocols on the land treadmill and underwater treadmill according to etiology.

		Land Treadmill	UWTM
**Extrusion IVDD**	**Post-surgical subacute dogs** [64]	**Speeds:** 1–2.5 km/h **Duration:** 20–30 min **Repetitions:** 2–3 times/day, 6 days/week **Slope:** 10 to 25°	**Speeds:** 1st day ≤ 0.8 km/h. progressively increasing to 1.2, 1.9, and 2.5 km/h. **Duration:** first for 3–10 min and extending to 20 min **Repetitions:** once a day within the first 2 weeks **Water temperature:** 24 and 26 °C **Water level:** between the lateral malleolus of the tibia and the lateral condyle of the femur
	**Post-surgical acute dogs** [19]	**Speeds:** 0.8 km/h (0.5 mph) to a maximum of 1.9 km/h (1.2 mph). **Duration:** start 5 min with the aim of reaching 20 min. **Repetitions:** start 4–6 times/day, 6 days/week with the aim of reaching 2 times/day, 6 days/week **Goal:** reach 30–40 min (2 to 3 times/day, 6 days/week). **Slope:** 10 to 25°	**1st:** 2–7 days after admission. **Speeds:** 1–3.5 km/h (2.2 mph) Duration: first for 5 min until reaching 1 h (5 days a week) **Repetitions:** once a day **Water temperature:** 26 °C
	**Post-surgical chronic dogs** [20]	**1st:** second day of admission. Quadrupedal step training **Speeds:** start 0.8 km/h (0.5 mph) with a maximum of 1.9 km/h (1.2 mph); **Duration:** start 5 min to achieve 20 min. **Repetitions:** start 4–6 times/day, 6 days/week to achieve 2–3 times/day, 6 days/week. **Goal:** reach 30–40 min (2 to 3 times/day, 5–6 days/week). **Slope:** 10 to 25°	**1st:** 48 h after admission **Speeds:** 1–3.5 km/h (2.2 mph) **Duration:** first for 5 min until reaching 1 h (5 days a week) **Repetitions:** once a day, 5 days/week **Water temperature:** 26 °C **Note:** Four sessions with good performance indicated a 10% increase in speed and duration.
		**Protocol:** **1st–2nd week:** 5–10 min, 0.8–1.9 km/h, 4–6 times/day, 6 days/week; **3rd–4th week:** 20 min, 2 km/h, 2–4 times/day, 6 days/week; **5th–6th week:** 30 min, 2.2 km/h, 2–3 times/day, 6 days/week; **7th–8th week:** 40 min, 2.5 km/h, 2 times/day, 6 days/week, 5° slope; **9th–10th week:** 40 min, 2.5 km/h, 2 times/day, 5 days/week, 10° slope; **11th–12th week:** 40 min, 2.5 km/h, 1 time/day, 5 days/week; 25° slope	**Protocol:** **1st–2nd week:** 5–10 min, 1–1.2 km/h; **3rd–4th week:** 10–20 min, 1.8–2 km/h; **5th–6th week:** 30 min, 2–2.5 km/h, 5° slope; **7th–8th week:** 40 min, 2.8–3 km/h, 5° slope; **9th–10th week:** 40 min, 3–3.5 km/h, 5° slope; **11th–12th week:** 60 min, 3.5 km/h, 10° slope
**Trauma**	**Spinal cord contusion cats** [18]	✓ BLT **Speeds:** started at 0.8 km/h (0.22 m/s) and increased to 1.2 km/h (0.33 m/s). **Duration:** started 2–5 min, increasing progressively to achieve 20 min **Repetitions:** started 3–6 times/day, to achieve 3 times/day; 6 days/week. **Slope:** without a slope.	**Duration:** first for 5 min until reaching 40 min **Repetitions:** once a day, 5 days/week. **Slope:** 10%. **Water temperature:** 24–26 °C
		✓ QLT Speeds: between 1 km/h (0.27 m/s) and 1.8 km/h (0.5 m/s), Duration: start 2–5 min, aiming sessions longer than 30 min Repetitions: start 4–8 times/day aiming sessions 3 times/day; 6 days/week. Slope: 10–25%	
	**ANNPE Dogs** [63]	**1st Phase:** ✓ DPN: **Speed:** 1.5 km/h **Duration:** 3–10 min **Repetitions:** 6–8 times/day; 6 days/week **Slope:** 2–5%	**1st Phase:** ✓ DPN: **Speeds:** 1.2–2 km/h **Duration:** 10–20 min **Repetitions:** 1 time/day; 5 days/week **Slope:** no slope
		✓ DPP: **Speed:** 1.8 km/h **Duration:** 3–10 min **Repetitions:** 4–6 times/day; 6 days/week **Slope:** no slope	✓ DPP: **Speeds:** 1.2–2 km/h **Duration:** 5–10 min **Repetitions:** 1 time/day; 5 days/week **Slope:** no slope
		**2nd Phase:** ✓ DPN: **Speeds:** 1.8–2.5 km/h **Duration:** 10–40 min **Repetitions:** 2–3 times/day; 5 days/week **Slope:** 2–5%	**2nd Phase:** ✓ DPN: **Speeds:** 2.8–4.5 km/h **Duration:** 40 min **Repetitions:** 1 time/day; 5 days/week **Slope:** 5–10%
		✓ DPP: **Speeds:** 2–2.5 km/h **Duration:** 10–40 min **Repetitions:** 2–3 times/day; 3 days/week **Slope:** 2–5%	✓ DPP: **Speeds:** 2–2.5 km/h **Duration:** 30 min **Repetitions:** 1 time/day; 3 days/week **Slope:** 2–5%
	**Cervical Extrusion IVDD post-surgical dogs** [62]	**Speeds:** starts 0.8–1 km/h **Duration:** Starts with 2–5 min and progressively increase up to 30 min. **Repetitions:** 4–6 times/day; 3–5 days/week **Slope:** no slope	**1st:** 48 h after admission **Speeds:** start 1.2 km/h **Duration:** first for 2–5 min until reaching 40 min **Repetitions:** once a day, 5 days/week. **Slope:** 10%. **Water temperature:** 24–26 °C **Water line:** near the tibial proximal epiphysis
		**Protocol:** **1st day:** 0.8–1 km/h, 2–5 min, 2–5 times **2nd day:** 0.8–1.2 km/h, 2–5 min, 3–5 times **3rd day:** 1 km/h, 5 min, 4–5 times **4th day:** 1.2 km/h, 5 min, 4 times **5th day:** 1.8 km/h, 5 min, 4 times **6th day:** 2 km/h, 10 min, 4 times **7th–12th day:** 2.5 km/h, 10 min, 3 times **13th–14th days:** 3 km/h, 15–30 min, 2 times	**Protocol:** **2nd–5th days:** 1.2 km/h, 2–5 min **6th day:** 1.5 km/h, 5 min **7th day:** 1.8 km/h, 5 min **8th day:** 2 km/h, 10 min **9th day:** 2 km/h, 15 min **10th–11th days:** 2.2 km/h, 15 min **12th–13th days:** 2.5 km/h, 15–30 min **14th–15th days:** 3 km/h, 30–40 min
	**Degenerative Myelopathy**	**Speeds:** start with 0.9–1.2 km/h, increasing to 1.2–2 km/h, following 2–2.8 km/h. **Duration:** start with 10–20 min, increasing to 20–30 min, following 30–40 min. **Repetitions:** 3 times/day; 6 day/week. **Goal:** 2.8–3.2 km/h, 60 min, 2 times/day, slope 10%.	**Speeds:** start with 0.9–1.2 km/h, increasing to 1.2–2 km/h, following 2–2.8 km/h. **Duration:** 10 min, increasing to 20 min, following 30 min. **Repetitions:** 1 time/day; 5 days/week. **Goal:** 2.8–3.2 km/h, 60 min, 2 times/day, slope 5–10%. **Water line:** lateral condyle of the femur.
		**Protocol:** **1st week:** 0.9–1.2 km/h; 10–20 min; 5 times/day **2nd week:** 1.2–2 km/h; 20–30 min; 4 times/day **3rd week:** 2–2.8 km/h; 30–40 min; 3 times/day **4th week:** 2.8–3.2 km/h; 40–60 min; 2 times/day	**Protocol:** **1st week:** 0.9–1.2 km/h; 10 min **2nd week:** 1.2–2 km/h; 20 min **3rd week:** 2–2.8 km/h; 30 min **4th week:** 2.8–3.2 km/h; 60 min

**Legend:** UWTM (underwater treadmill); IVDD (intervertebral disc disease); BLT (bipedal locomotor training); QLT (quadrupedal locomotor training); DPN (deep pain negative); DPP (deep pain positive); ANNPE (acute non-compressive nucleus pulposus extrusion).

**Table 5 animals-12-03582-t005:** Key point n° 4—Locomotor training in deep pain positive and deep pain negative patients.

**Keypoint n° 4:** *“Locomotor training is essential manly in subacute dogs and cats deep pain negative (DPN) or deep pain positive (DPP) of grade 1 (according to the modified Frankel scale).”*

## Data Availability

The data presented in this study are available upon request from the corresponding author.

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
