# Peer review of "Approach to Small Animal Neurorehabilitation by Locomotor Training: An Update"

_animals, 2022, doi:10.3390/ani12243582_

Round 1

Reviewer 1 Report

General comments:

This review aimed to provide an overview of the current evidence on the locomotor training approach to gait rehabilitation after SCI.

The primary guidelines on locomotor training provided in this manuscript for small animals recovering from SCI will be meaningful in everyday life; can be widely applied in veterinary clinics and clinical trials.

Specific comments:

  1. Line 2: in the tittle, the authors should use “small animal” instead of “veterinary”.
  2. Line 105: please rephrase “…to induce leg muscle electromyography…”.
  3. Line115: “low-level” instead of “low-lever”.
  4. Lines 18-119: please provide more information about the way to choose the correct device to promote locomotion following SCI.
  5. Line 121: also “interlimb coordination”…?
  6. Lines 130-131: I do not understand the values “82%” and “18%” about the numbers of daily sessions. What is the impact from a translational perspective (from rats to small animals)? It is important to have references from companion animal’s studies.
  7. Line159: “soft music” is similar to “classical music”?
  8. Table 4 and along 3.1: I wonder why the Land Treadmill is applied 6 days/week and the UWTM just 5 days/week?
  9. Line 245: about muscle fatigue, the authors could provide some studies done on dogs/cats with the addition of weight-bearing pads, providing the main benefits.
  10. Line 291: please provide more information about the anti-inflammatory role of the locomotor training, adding more references.

Reviewer 2 Report

This interesting research, needs some clarification before being considered for publication. My minor comments: please remove the dot at the end of the title. The Simple Summary section is too laconically written.Introduction section: Functional neurorehabilitation (FNR) is based on neuroanatomy and neurophysiology principles, allowing intensive protocols to be implemented through specific guidelines in human medicine-please clarify this sentence, neuroanatomy and neurophysiology principles-meaning what specifically??, allowing intensive protocols to be implemented through specific guidelines-what did the authors mean specifically????.I have a question for land treadmill (bipedal step training on a cat), does this presented board on which the cat rests its front paws does not cause the cat too much stress? does the slippery surface of this board does not cause that the cat has to excessively mobilize the work of muscles to keep on this board? to maintain balance? For Review Articles data should be collected according to current guidelines and checklists (e.g., PRISMA: Preferred Reporting Items for Systematic Reviews and Meta-Analyses).
